# `SILENCE`: Lightweight Protection for Privacy in Offloaded Speech Understanding

**Dongqi Cai**
Beiyou Shenzhen Institute
Shenzhen, Guangdong
cdq@bupt.edu.cn

**Shangguang Wang**
Beiyou Shenzhen Institute
Shenzhen, Guangdong
sgwang@bupt.edu.cn

**Zeling Zhang**
Beiyou Shenzhen Institute
Shenzhen, Guangdong
marovlo@bupt.edu.cn

**Felix Xiaozhu Lin**
University of Virginia
Charlottesville, VA, 22904
felixlin@virginia.edu

**Mengwei Xu**
Beiyou Shenzhen Institute
Shenzhen, Guangdong
mwx@bupt.edu.cn

## Abstract

Speech serves as a ubiquitous input interface for embedded mobile devices. Cloud-based solutions, while offering powerful speech understanding services, raise significant concerns regarding user privacy. To address this, disentanglement-based encoders have been proposed to remove sensitive information from speech signals without compromising the speech understanding functionality. However, these encoders demand high memory usage and computation complexity, making them impractical for resource-constrained wimpy devices. Our solution is based on a key observation that speech understanding hinges on long-term dependency knowledge of the entire utterance, in contrast to privacy-sensitive elements that are short-term dependent. Exploiting this observation, we propose `SILENCE`, a lightweight system that selectively obscuring short-term details, without damaging the long-term dependent speech understanding performance. The crucial part of `SILENCE` is a differential mask generator derived from interpretable learning to automatically configure the masking process. We have implemented `SILENCE` on the STM32H7 microcontroller and evaluate its efficacy under different attacking scenarios. Our results demonstrate that `SILENCE` offers speech understanding performance and privacy protection capacity comparable to existing encoders, while achieving up to $53.3\times$ speedup and $134.1\times$ reduction in memory footprint.

## 1   Introduction

**Privacy concern for cloud speech service**  The volume of speech data uploaded to the cloud for spoken language understanding (SLU) is steadily increasing [1, 2, 3], particularly in ubiquitous wimpy devices where textual input is inconvenient [4, 5, 6], e.g., home automation devices [7], smartwatches [8], telehealth sensors [9] and smart factory sensors [10] . However, exposing raw speech signal to the cloud raises privacy concerns [11]. It was revealed that contractors regularly listened to confidential details in Siri recordings to improve its accuracy [12]. This included private discussions, medical information, and even intimate moments.

There are many aspects of potential privacy leakage in cloud-based SLU. Among them: biometric or contextual privacy leakage have been well studied and somewhat solved by removing information relevant to such tasks without compromising the SLU accuracy [13, 14]; transcript protection (especially sensitive entities) is more challenging since it is deeply entangled with the SLU task itself. As

38th Conference on Neural Information Processing Systems (NeurIPS 2024).

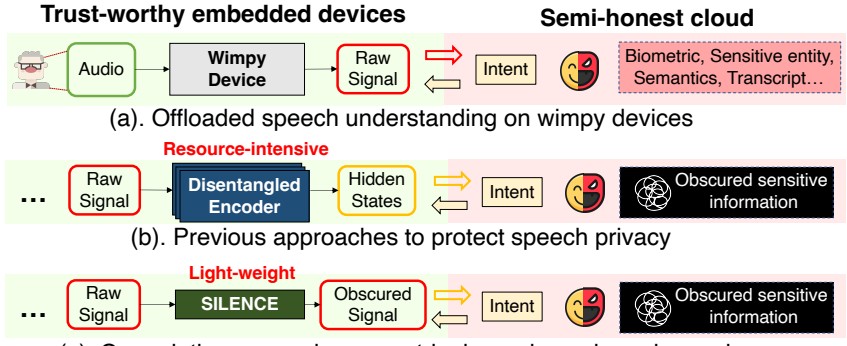

Figure 1: Illustration of offloaded speech understanding on resource-constrained devices and its privacy protection.

shown in Figure 1, this paper focus on ensuring that cloud-based systems could efficiently classify the intent of SLU task (e.g., scheduling appointments or controlling home devices) while refraining from identifying the concrete entities (e.g., unintended names or passwords) in the spoken utterance, i.e., high word error rate (WER) of Automatic Speech Recognition (ASR) task. This is also a setting commonly used in speech privacy protection [15, 16, 17, 11, 18].

**Prior approaches** A prevalent method for private speech processing is employing *encoders*[1] based on disentanglement representation learning [15, 16, 19, 20], as illustrated in Figure 1(b). Those encoders extract the speech representations using pre-trained acoustic models, e.g., wav2vec [21, 16], conformer [22, 20] and Preformer [23, 15]. Furthermore, they promote representation disentanglement through adversarial training [24]. For example, PPSLU [15] uses a 12-layer transformer-based Preformer as its encoder.

As a result, disentanglement-based encoders still demand considerable computational resources, often exceeding tens of GFLOPs, to achieve effective disentanglement [25]. They are also memory-intensive, often comprising tens of millions of parameters. Consequently, they are unsuitable for embedded devices with limited memory. Moreover, it takes time-consuming adversarial training to disentangle the encoded representation for each specific SLU task. This aspect limits the flexibility and scalability for emerging SLU tasks. More motivating details will be presented in §2.2.

In this paper, we aim to achieve the real-time, privacy-preserving offloading of speech understanding task on wimpy devices like STM32H7 microcontroller [26] with only 1MB RAM. This goal necessitates a novel encoder design that must be both lightweight and effective in filtering out sensitive information, as illustrated in Figure 1(c).

**Our solution** We therefore present SILENCE, a **SI**mp**L**e **ENC**od**E**r designed for efficient privacy-preserving SLU offloading. It is based on the *asymmetric dependency* observation: SLU intent extraction (e.g., scenario identification) typically requires only long-term dependency knowledge across the entire utterance, while ASR task (e.g., recognizing individual words or phrases) needs short-term dependency, as confirmed by our experiments in §3.1. Based on it, SILENCE strategically partitions the utterance into several segments, selectively masking out the majority to enhance privacy by obscuring short-term details, without significantly damaging the long-term dependencies. The processed audio waveform is then transmitted to the cloud for SLU intent analysis. Additionally, we integrate a differential mask generator, inspired by interpretable learning methods [27], to optimize performance by automatically identifying how many and which segments to mask.

**Results** We deploy SILENCE on the STM32H7 microcontroller [26] and assess its performance using the SLURP dataset [28] in both black-box and white-box attack environments. SILENCE achieves 81.2% intent classification accuracy on SLURP, surpassing previous privacy-preserving SLU systems by up to 8.3%. Regarding privacy protection, SILENCE offers comparable security

---

[1]Note that these encoders are not specifically transformer encoders; rather, they can be implemented using any NNs to encode speech signals.

to earlier systems, with a word error rate of up to 81.6% and an entity error rate of 90.7% under malicious ASR attacks. Even against white-box attacks, where attackers are strongly assumed to have the same encoder structure and weights as `SILENCE`, plus partial data from malicious clients, `SILENCE` maintains 67.3% word error rate and 64.3% entity error rate. Additionally, `SILENCE` proves to be resource-efficient and feasible for wimpy devices, using only 394.9KB of memory and taking just 912.0ms to encode a 4-second speech signal. Integrated with RPI-4B for a fair comparison, `SILENCE` uses up to $134.1\times$ less memory and operates up to $53.3\times$ faster than prior systems. The accuracy of `SILENCE` is only 7% lower than unprotected SLU systems.

**Contribution** We have made the following contributions.

- Based on the observation of asymmetric dependency between SLU and ASR tasks, we propose `SILENCE`, a simple yet effective encoder system for privacy-preserving SLU offloading.
- We are the first to retrofit interpretable learning methods to automatically configure the masking process for a better balance between privacy and utility in speech understanding tasks.
- We evaluate `SILENCE` on a wimpy microcontroller unit and demonstrate its effectiveness under various attack scenarios.

## 2 Related Work and Background

### 2.1 Privacy-preserving SLU

Spoken Language Understanding (SLU) is a critical component of modern voice-activated systems, responsible for interpreting human speech and translating it into structured, actionable commands. For instance, when a user says, "Set a meeting for tomorrow at 10 AM," the SLU system might map this to a structured intent such as {scenario: Calendar, action: Create_entry}. Long-dependent intend classification is currently the main objective of SLU understanding literature and has a wide range of application scenarios [29, 30, 31, 32, 33, 28, 34, 35, 36].

**Evolution of SLU Systems** The evolution of SLU systems has seen a shift from traditional two-component systems, comprising ASR and Natural Language Understanding (NLU), to modern end-to-end neural networks [37, 38]. These advanced systems bypass the intermediate textual representation and directly map speech signals to their semantic meaning, enhancing efficiency and reducing error propagation. A typical end-to-end SLU model features an encoder, often with convolution and attention-based elements, and a decoder, including a transformer decoder and a connectionist temporal classification decoder. Many SLU systems incorporate encoders from pre-trained ASR models like HuBERT [39], replacing the original ASR decoder with one tailored for SLU tasks.

**Threat Model** Our threat model aligns with prior work [15, 16] where users (the victims) actively offloads their audio data to the cloud server (the adversary) for intended SLU tasks. Upon receiving the data, the adversary may employ automatic speech recognition to transcribe the audio and identify private entities [17, 11, 18]. Note that the transcriptions are often exceedingly detailed, containing much more information than the users intend to disclose. The goal of this paper is to ensure that the victims can reliably obtain the predefined SLU intent from the adversary, while preserving the adversary from discerning sensitive details or private entities in the transcript.

For instance, home pods might capture recordings of confidential daily interactions alongside explicit commands, presenting a paradigmatic case for `SILENCE`. Without `SILENCE`, over 80% of our private daily conversations could be automatically recognized and stored for unforeseen usage as will be analyzed in §5.1.

### 2.2 Inefficiency of Existing Approaches

**Privacy-preserving methods** Crypto-based approaches, such as HE [40] and MPC [41], have been proposed to provide encrypted computation. Unfortunately, they are technically slow and thus impractical for deployment on resource-constrained audio devices due to the significant increase in computation and communication complexity. For example, MPC-based PUMA [42] takes 5 min-

utes to complete one token inference, which is far too slow for real-time. Voice conversion is another method to protect speech content. Prɛɛch [43] integrates voice conversion with GPT-based generated noise protect privacy, but it is far from feasible for deployment on wimpy devices. Traditional peripheral devices, such as ultrasonic microphone jammers (UMJ), are designed to obscure raw speech by inserting non-linearity noise, thereby preventing illegal eavesdropping[44, 18]; however, they also corrupt speech semantics as well. A emerging and prevailing strategy is disentangling-based encoders [16, 15, 19]; they aim to create a disentangled and hierarchical representation of the speech signal devoid of sensitive data. But we reveal their performance issue next.

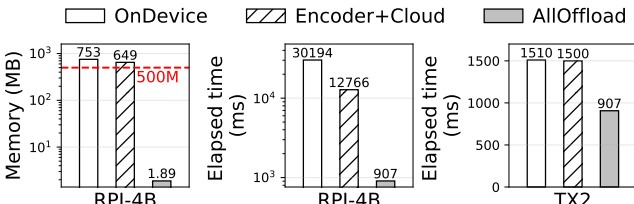

Figure 2: Cost of disentangling-based encoders [15] for a 4-second audio inference.

We conduct preliminary experiments to measure the resource consumption of the disentangling-based encoder of a pre-trained SLU model on a Raspberry Pi 4B (RPI-4B) [45] and Jetson TX2 (TX2) [46]. Our key observation is that disentangling-based privacy-preserving SLU system is too resource-intensive for practical deployment. As illustrated in Figure 2, a disentanglement encoder consumes 648.7MB memory and 12.8s for complete one inference on RPI-4B. Even in the strong TX2 with GPU, the encoder still takes 593.0ms to complete one inference. Considering the network latency, the end-to-end latency of the disentangling-based SLU offloading system only saves 0.7% wall-clock time compared to the `OnDevice` inference without offloading, with a similar memory footprint over 500M.

***Implications*** Disentangling-based encoders is slow and memory-intensive due to the complex encoder structure designed to separate sensitive information from the speech signal. Given the limited resource of wimpy devices, it is not practical for common privacy-preserving SLU scenarios. To enable practical privacy-preserving SLU, the encoder structure and the inference process need to be simplified.

## 3 `SILENCE` Design

### 3.1 System Design and Rationales

We introduce `SILENCE` to efficiently scrub raw audio for privacy-preserving SLU, as depicted in Figure 3. The key idea of `SILENCE` is simple and novel: it masks out a portion of audio segments before sending them to the cloud for SLU tasks. This design is based on an unique observation shown in Figure 4(c): when a portion of audio segments is masked out, the ASR model becomes incapable to recognize the phonemes in the masked frames, while the SLU model can still recognize the intent.

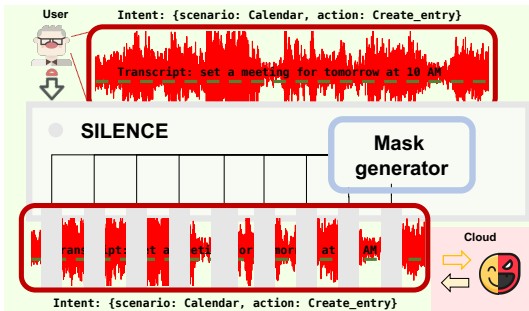

Figure 3: `SILENCE` overview. Red hard line represents the long-term dependency, while the green dotted line represents the short-term dependency.

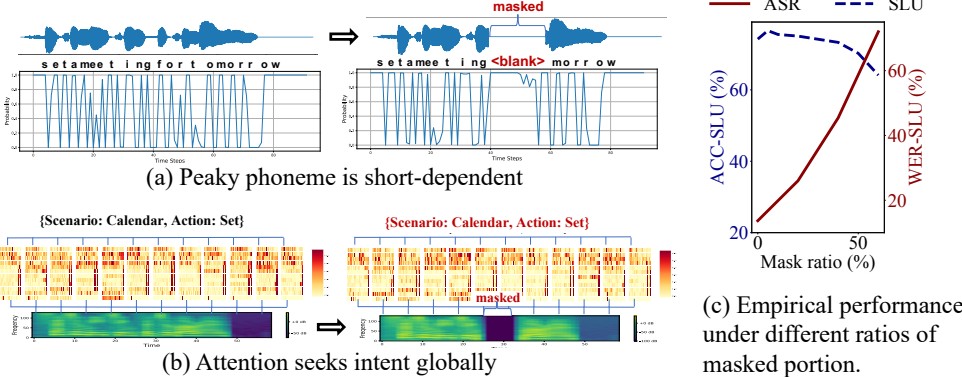

(a) Peaky phoneme is short-dependent

(b) Attention seeks intent globally

(c) Empirical performance under different ratios of masked portion.

Figure 4: Foundation of SILENCE: asymmetrical dependency. (a). ASR task is short-term dependent on the peaky phoneme probability. (b). SLU task is long-term dependent on knowledge from the whole utterance. (c). Empirical results.

**Design rationale** Why is SILENCE able to protect the sensitive entity privacy while maintaining SLU accuracy? This capability is rooted in the *asymmetrical dependency* between the ASR and SLU task.

Speech is composed of many meta phonemes, and the generation of a single meta phoneme depends on its adjacent frame [11]. *Dependency* is defined as the length of frame that a model's output depends on. Figure 4(a) shows each phoneme is mainly dependent on a few frames, indicating short-term dependency. This phenomenon is referred to as "peaky behavior" in the ASR literature [47]. In contrast, an SLU model utilizes an attention-based decoder [39] to capture the relationship between the entire utterance and the intent, implying that the intent is long-term dependent on the whole utterance.

Formally, SILENCE is a simple encoder based on asymmetrical dependency-based masking. This simple masking encoder is defined as: $\hat{x} = x \odot \mathbb{Z}$, where $x$ is the input audio signal, $\odot$ represents the element-wise multiplication, $\hat{x}$ is the masked audio signal and $\mathbb{Z}$ is the binary masking vector with the same dimension as $x$. $\mathbb{Z}$ consists of $k$ uniform portion, with all 0s or 1s in one portion to mask-out or preserve the complete adjacent frames, respectively. This simple encoder forms the basis of SILENCE's efficiency and privacy-preservation capacity, enabling secure offloading of speech understanding tasks on wimpy devices.

**The configuration challenges:** Figure 4(c) demonstrates that the ratio of masked portion plays a crucial role in balancing the privacy (WER-ASR) and utility (ACC-SLU). Currently, SILENCE employs a trivial masking mechanism, necessitating clients to undertake a time-intensive hyper-parameter adjustment about the extent and location of masking. Incorrect masking configurations can result in significant loss of global long-term dependency, negatively affecting SLU accuracy, or insufficient masking of sensitive information, thus compromising privacy. Therefore, we face critical questions: how many and which portions should be masked?

### 3.2 Online Configurator for SILENCE

To address these challenges, we derive a differential mask generator from the interpretable learning [27] as a online configurator for SILENCE. This automatically generate the masking vector $\mathbb{Z}$. The mask generator is trained to identify how many and which portions to mask, optimizing the privacy-utility balance.

**Differentiable mask generator** The configurator model aims to minimize the discrepancy between masked and original output by generating a mask $\mathbb{Z}$. Formally, we define the number of unmasked portions as $\mathcal{L}_0$ loss:

$$\mathcal{L}_0(\phi, x) = \sum_{i=1}^{n} \mathbf{1}_{[\mathbb{R}_{\neq 0}]}(\mathbb{Z}_i) \tag{1}$$

where $\phi$ is the mask generator, $\mathbf{1}(\cdot)$ is the indicator function. We minimize $\mathcal{L}_0$ for dataset $\mathcal{D}$, ensuring that predictions from masked inputs resemble those from the origin model:

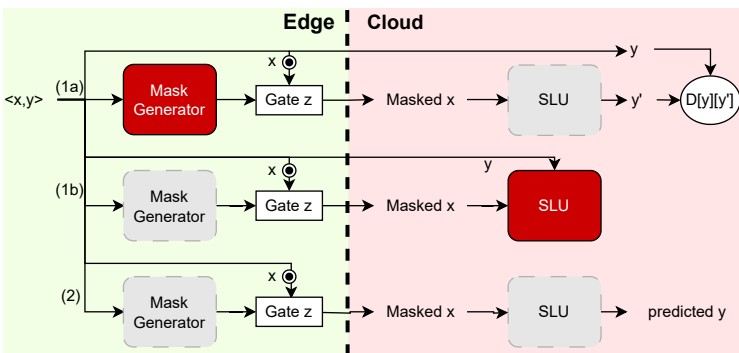

Figure 5: SILENCE workflow. (1) *Offline phase*: (**1a**) Training mask generator and (**1b**) adapting cloud SLU model to it; (2) *Online phase*: Conducting could inference with the masked x. Only masked input audio x and insensitive intent label y are exposed to the cloud.

$$\min_{\phi} \sum_{x \in \mathcal{D}} \mathcal{L}_0(\phi, x) \tag{2}$$

$$\text{s.t. } D_{\star}[y\|\hat{y}] \leq \gamma \quad \forall x \in \mathcal{D} \tag{3}$$

where $\hat{y} = f(\hat{x})$, $y$ is the tokenized label, $D_{\star}[y\|\hat{y}]$ is the KL divergence and the margin $\gamma \in \mathbb{R}_{>0}$ is a hyperparameter.

Given that $\mathcal{L}_0$ is discontinuous and has zero derivative almost everywhere, and the mask generator $\phi$ requires a discontinuous output activation (like a step function) for binary masks, we utilize a sparse relaxation to binary variables [48, 49] instead of the binary mask during training.

**Holistic workflow** As shown in Figure 5, SILENCE encompasses two phases:

(1) *Offline phase*: (**1a**) First, SILENCE trains a differentiable mask generator. The client selects a mask generator model, potentially a submodule of a pre-trained ASR model, such as HuBERT's CNN feature extractor. A small gate model is then integrated with this submodule. The combined model processes the input audio and generates a mask. This mask selectively conceals parts of the input, ensuring retention of only vital SLU information while hiding sensitive data. The masked input is then forwarded to either a trusted cloud service or a local SLU model for obtaining masked output. The mask generator is fine-tuned to minimize the discrepancy between the masked output logits and the original intent, as defined in Equation (1-3).

(**1b**) Second, SILENCE adapts the cloud model . Here, the client forwards the masked input and a specific SLU intent (e.g., "set alarm") to the cloud-based SLU model. The model undergoes fine-tuning to adapt to the masked inputs. This process includes adjusting the model parameters for accurate recognition and response to SLU commands based on the masked input.

(2) *Online phase*: In online speech understanding, the client sends the masked input to the cloud SLU model. Using the adapted model, the cloud-based SLU accurately identifies and executes the intended SLU action or response.

**Configurator cost analysis** Training the differentiable mask generator is affordable for the client. Our experiments indicate that convergence is achieved with approximately 200 audio samples, equivalent to 600 seconds of audio. This process takes up to 30 seconds on an A40 GPU. Adapting the SLU model to each mask generator is a one-pass effort. This adaptation is relatively trivial, especially when starting from a fine-tuned SLU model rather than building from scratch. This aspect of the process incurs minimal cost compared to the training of the cloud SLU model. Moreover, these costs can be amortized over a large number of edge users in the long run, making it an economically viable solution.

**Remark** Note that the mask generator is not developed for tagging sequences at a semantic level. Rather, its design focuses on identifying segments that are more relevant to the SLU task. This task is essentially a relatively straightforward binary classification problem, which is proven to be effective in prior interpretable learning literature [27, 49] and light-weight enough for real-time inference.

# 4 Implementation and Methodology

We have fully implemented the SILENCE prototype atop SpeechBrain [50], a PyTorch-based and unified speech toolkit. As prior work [39], we use SpeechBrain to train the differential mask generator and simulate the cloud training process. After that, we deploy the trained mask generator into the embedded devices and evaluate the end-to-end performance.

**Hardware and environment** Offline training is simulated on a server with 8 NVIDIA A40 GPUs. The trained mask generator is deployed into the STM32H7 [26] or Raspberry PI 4 (RPI-4B) [45]. STM32H7 is a resource-constrained microcontroller with 1MB RAM. RPI-4B is a popular development board with 4GB RAM. We embed the approaches not feasible to fit in the STM32H7 into the RPI-4B.

**Models** We design four types of mask generator structures: (1) Random: a random binary vector generator with 50% portion masked; (2) SILENCE-S: a learnable mask generator with only one MLP gate; (3) SILENCE-M: a learnable mask generator with one HuBERT encoder layer and the gate; (4) SILENCE-L: a learnable mask generator with three HuBERT encoder layers and the gate. As for the cloud SLU model, we simulate it using the SoTA end-to-end SLU model [39]. It replaces the ASR decoder of pre-trained HuBERT with SLU attentional decoder.

**Dataset and Metrics** We run our experiments on SLURP [28] and FSC [51]. FSC is a widely used dataset for spoken language understanding research. SLURP's utterances are complex and closer to daily human speech, We select scenario classification accuracy to measure the SLU understanding performance (ACC-SLU). Following prior work [15], we choose large-scale English reading corpus LibriSpeech [52] for a multi-task protection scenario. In the multi-task protection scenario, not only the SLU command utterance (SLURP/FSC) but also the background or the subsequent utterance (LibriSpeech) are uploaded to the cloud. WER is used to measure the attack performance. More specifically, we utilize WER-SLU to measure the attacker's capacity to recognize the word information in the uploaded SLU audio itself, and WER-ASR as the WER of recognized accompanying audio, i.e., LibriSpeech dataset. We also report the private entity recognition error rate (EER) to ensure that the cloud model is not able to recognize the private information in the speech signal. As for latency, we sequentially fed test audios into the local model without any window processing[2] and recorded the average forward time as the local execution time.

**Baselines** We compare SILENCE to the following alternatives: (1) OnDevice means the cloud SLU model is downloaded and run locally on the client device. (2) AllOffload means the raw audio is uploaded to the cloud for SLU inference. (3) VAE [16] is the vanilla variational auto-encoder method that uses adversarial training to disentangle the private information from speech signal. (4) PPSLU [15] is the state-of-the-art disentangling-based SLU privacy-preserving system, which uses 12 transformer layers to separate the SLU information into a part of the hidden layer and only sends those hidden layers to the cloud for SLU inference.

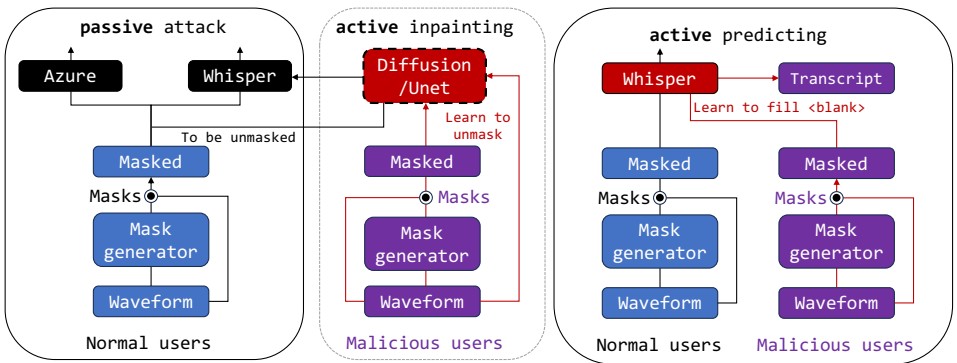

Figure 6: Mask generator and different attack scenarios, including both passive and active attacks.

---

[2]The average duration of test SLU snippets is 2.8 seconds, with a maximum of 21.5 seconds, which is shorter than the maximum input window of speech models (e.g., 30 seconds for Whisper [53]).

**Attack scenarios.** As illustrated in Figrue 6, we use five attacks encompassing both active and passive attacks: (1) `Azure` represents a passive black-box attacker scenario, in which the masked audio is transmitted to Azure [54] for automatic speech recognition. (2) `Whisper` simulates a SoTA cloud-based ASR model. This passive black-box attacker uses the pre-trained $Whisper.medium.en$ model [53], directly downloaded from HuggingFace [55]. (3) `Whisper(White-box)` constitutes an active white-box attack. Here, we hypothesize that certain users are malicious and disclose the mask generator's structure and weights, along with their own audio data, to the `Whisper` attack model. `Whisper(White-box)` then utilizes this collected data from malicious users to adapt the pre-trained $Whisper.medium.en$ model to the specific masking pattern. (4) `U-Net` is a traditional inpainting model based on convolutional U-Net structure, commonly used in literature to actively reconstruct missing audio signals [56, 57]. We utilize the SLURP training set and their masked counterparts to train the inpainting model from scratch to reconstruct the missing audio. (5) `CQT-Diff` is a neural diffusion model with an invertible Constant-Q Transform to leverage pitch-equivariant symmetries [58], allowing it to effectively reconstruct audio without retraining.

**Hyper-parameters** During the offline phase in Figure 5, we use the Adam optimizer with a learning rate of 1e-5 and a batch size of 4. For the inference step, we use the batch size of 1 to simulate the real streaming audio input scenario. The end-to-end cloud SLU latency is measured by invoking Azure APIs following previous work [59]. KL threshold $\lambda$ is set as 0.15 for all mask generators. Attack model is set as `Whisper` without special declaration. We have an illustrative example of the generated masks on audios selected randomly from SLURP in supplimentary material.

## 5  Evaluation

### 5.1  End-to-end performance

`SILENCE` **achieves comparable accuracy performance and privacy protection capacity to previous encoders.** As shown in Figure 7, we compare the accuracy of `SILENCE` with all baselines. `OnDevice` offloads no signals to the cloud and thus has the best privacy protection (WER=100). It is observed that `SILENCE` could achieve up to 81.1% accuracy, with less than 7% accuracy loss compared to unprotected `AllOffload` and local `OnDevice` SLU model. Its rationale is that we mainly mask the short-dependent frames that does not significantly affect the SLU performance. We also compare the performance of `SILENCE` with the SoTA privacy-preserving SLU system, i.e., `PPSLU` [15]. `SILENCE` achieves 7.2% higher accuracy than `PPSLU` which tries to apply complex non-linear transformation to the hidden layer to prevent malicious re-construction, but this might also damage part of the SLU information. In terms of privacy preservation, our learnable mask generator achieves up to 78.6% WER using `SILENCE-L`, indicating a privacy-preserving capacity on par with `PPSLU`. The same benefits exist in FSC dataset as well. `SILENCE` demonstrates more than 99% intent understanding accuracy, similar to all the baselines, while effectively defends against sensitive word recognition attacks, achieving more than 80% WER, outperforming all disentanglement-based protections. Furthermore, we complete the inference with much lower delays and memory footprint as will be shown in Figure 10.

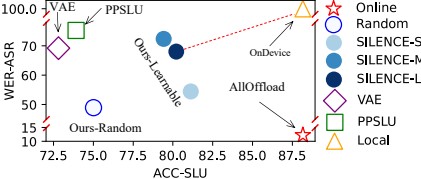

Figure 7: Performance of different privacy-preserving SLU approaches.

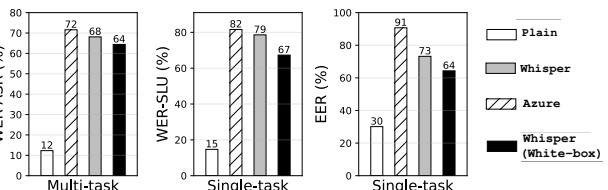

Figure 8: `SILENCE` privacy-preserving capacity under different attack models.

`SILENCE` **is resistant to different attack models.** As illustrated in Figure 8, `SILENCE` increases the SLU-WER from 14.7% to 78.6% under the attack model `Whisper`. As for the online attack model `Azure`, `SILENCE` increases the SLU-WER from 14.7% to 81.6%. According to our returned service details, we find that over 50% of the sent audios are tagged as "$ResultReason.NoMatch$", which means audios are recognized as null utterances by the Azure ASR model. `Whisper(White-box)` is a white-box attack model, which means the attacker has the same mask generator structure and

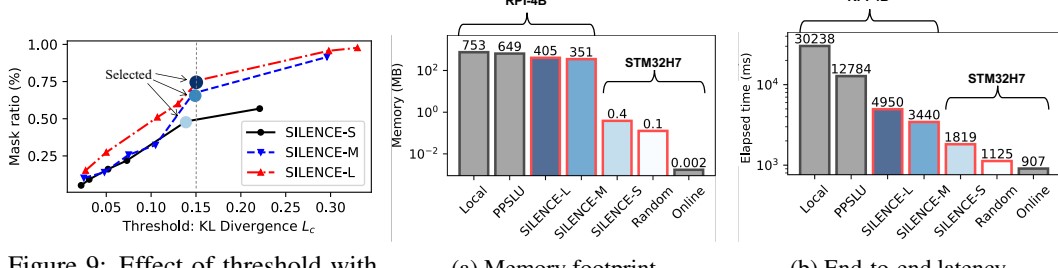

Figure 9: Effect of threshold with different mask generators

(a) Memory footprint

(b) End-to-end latency

Figure 10: Comparison of resource cost in different SLU approaches. Ours are highlighted in red.

weights as the SILENCE. We still achieve more than 50% SLU-WER under this attack model. This is because even Whisper(White-box) is fine-tuned to fill some of the missing frames, it still could not recover the private missing frames. Because masking the short-dependent frames fundamentally destroys the raw audio signal. It is not possible to re-construct the phoneme without knowing any speech information. In the last subfigure, we show the high entity error rate to demonstrate that the private entity is not leaked.

SILENCE can defend against the active inpainting attack as well. As shown in Table 1, U-Net can rarely reconstruct the masked audio. Even worse, it introduces incorrect noisy signals, degrading the attack success rate. CQT-Diff inpainting can fill the missing waveforms but cannot successfully reconstruct the content because it is designed to reconstruct background music, such as piano concertos. SLU audio, which includes human intent and conversation, is difficult to reconstruct.

|  | PlainText | Azure | Naive Whisper | U-Net | CQT-Diff | Whisper predict (white box) |
|---|---|---|---|---|---|---|
| WER-SLU (%) | 14.7 | 81.6 | 78.6 | 82.5 | 74.3 | 67.3 |
| WER-ASR (%) | 12.3 | 71.6 | 681. | 71.4 | 65.9 | 64.4 |

Table 1: Potential attack Word Error Rate (WER) under different attack scenarios.

SILENCE **scales to better privacy-accuracy trade-off with a larger mask generator.** We explore the impact of the threshold $\gamma$ of SILENCE under different mask generator structures. As shown in Figure 9, the threshold $\gamma$ controls the trade-off between the privacy and utility. When $\gamma$ is small, the mask generator is more conservative, leading to higher the utility a lower the masking portion. As we have discussed in Section 3, a lower rate of masking portions leads to higher possibility of privacy entity leakage. When $\gamma$ is large, the mask generator is more aggressive, enhancing privacy. Another way to achieve more practical privacy-utility balance is using a more complex mask generator structure, e.g., SILENCE-L. It achieves higher utility with the same privacy level compared to SILENCE-S, albeit with less efficiency, as shown in § 5.2.

## 5.2 System cost

SILENCE protects the private entities efficiently as shown in Figure 10. Different from prior encoders using complex disentanglement model, SILENCE only requires a light-weight mask generator to scrub the private information. The size of this generator varies according to different mask generator structures. For the smallest mask generator, SILENCE-S, it only requires a 394.9KB memory footprint, and could successfully embed into the resource-constrained STM32H7 with 2MB RAM. SILENCE is efficient not only in terms of memory footprint but also in latency. SILENCE-S completes the local encoding with only 912.2ms on the resource-constrained STM32H7. For a fair comparison, we embed SILENCE-S into RPI-4B and find that it is $18.1\times$ faster and $134.1\times$ less memory footprint than PPSLU. Even with the strong mask generator SILENCE-L, SILENCE achieves up to $7.5\times$ lower encoding latency and consumes $1.9\times$ less memory compared to OnDevice.

# 6 Conclusion and Discussion

SILENCE is an efficient and privacy-preserving end-to-end SLU system based on the asymmetrical dependency between ASR and SLU. SILENCE selectively mask the short-dependent sensitive words while retaining the long-dependent SLU intents. Together with the differentiable mask generator, SILENCE shows superior end-to-end inference speedup and privacy protection under different attack scenarios.

**Limitations:** While for the first time, SILENCE provides a feasible privacy-preserving solution for resource-constrained audio devices, it introduces a huge design space for mask generator structures. The mask generator is akin to a lock; a genius lock design can protect privacy in the smallest of spaces, but a poor lock design can be bulky and easily broken. In this work, we simply inherit the SLU model structure and instantiate three sub-models from it to demonstrate better efficiency than previous encoders. Researchers can explore other structures for a better privacy-accuracy-efficiency trade-off. We will open-source all the code and checkpoints to facilitate further research in this direction.

Some other potential limitations about lossy privacy-preserving capacity, the need for fine-tuning the cloud SLU model, the scope of defended threat model and the extension to offline scenarios are thoroughly discussed below for further clarification.

**Is current privacy-preserving capacity enough?** The quantitative WER 80% is considered secure enough, as previous encoders have strived to reach that level [15, 16]. And some SLU transcripts contain the intent word, so the successfully inferred word might be a non-private intent word. For instance, in one test audio transcript, "I want some jazz music to play", the intent is 'scenario': 'play', 'action': 'music'. The interpretation of the malicious cloud ASR, "all subjects were used to play", is acceptable since the predicted phrase "to play" contains no private information. This scenario is typical for most audios; we managed to preserve 90% of the private entities in Figure 7. This achievement matches the SoTA in privacy-preserving capacity, with up to $30\times$ lower latency and $100\times$ memory reduction.

**Why and how to fine-tune the cloud SLU Model?** Initially, the cloud SLU is a generic pre-trained speech model lacking the capability to accurately understand personalized user intent. It is crucial to fine-tune the cloud SLU for better personalized intent understanding[3]. Secondly, while short-dependent masking does not eliminate intent information, it does impact specific details within the attention map, as depicted in Figure 4(b). Fine-tuning the cloud SLU model helps mitigate this impact and enhances the understanding of the user's intent. Currently, cloud service providers have already offered APIs that allow users to fine-tune their personalized cloud speech model [60].

**Could private semantic detection attack be prevented?** We clarified that detecting short-dependent key phrases or specific commands is not the focus of this work. For example, eavesdropping on specific financial words and political framing are *out-of-scope*. However, we can offer defense capabilities against them. The mask generator, controlled by the user, is trained to scrub utterances unrelated to the public intent. Private entities not predefined by the user are almost never included in the masked audio. Therefore, even if an attacker possesses a well-defined semantic and the mask generator, training the detection threat model is challenging because the synthetic masked audio lacks clear representations of the private semantic.

**Extesion to offline scenarios:** Offline conditions occur periodically for resource-constrained devices. SILENCE can be easily integrated into an orchestration of small on-device SLU models and robust cloud models. This orchestration has been officially adopted by many off-the-shelf products, such as Apple Intelligence in iOS 18. Our system remains indispensable in such circumstances because small on-device SLU models may not generate satisfactory intent understanding due to their restricted model size. Even when on-device SLU models produce correct intent understanding, they cannot always operate due to limited device energy. As a result, online procedures are still the main components of current SLU solutions. The on-device functionality can be used as an alternative in offline conditions. With our system, the cloud-based SLU component is both privacy-preserving and efficient.

---

[3]Note that a general speech model is sufficient for training the local mask generator in Figure 5 step (1a), as the focus is not on generating precise intent but rather on obtaining a coarse-grained distribution of numerical logits to facilitate mask generator training.

## Acknowledgments and Disclosure of Funding

The authors thank the anonymous reviewers for their insightful feedbacks. This work was supported by the National Key Research and Development Program of China under Grant No. 2021ZD0113001, the Youth Program of the National Natural Science Foundation of China under Grant No. 62102045, the National Science Fund for Distinguished Young Scholars under Grant No. 62425203, and the CCF-Sangfor "Yuanwang" Research Fund. Dongqi Cai was supported by the China Scholarship Council under Grant No. 202406470054.

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
