# OpenReview forum: "SILENCE: Protecting privacy in offloaded speech understanding on resource-constrained devices"
_NeurIPS.cc/2024/Conference — NeurIPS 2024 poster_

### Official Review · Reviewer_YHpd · 2024-07-06

**Soundness:** 2
**Presentation:** 3
**Contribution:** 2
**Rating:** 5
**Confidence:** 3

**Summary:**

This paper investigates the perturbation of audio signals to impair the accuracy of automatic speech recognition (ASR) models, thus enhancing privacy protection, while ensuring that spoken language understanding (SLU) models maintain high accuracy for interpreting user intentions. The key insight is that ASR models depend on local utterances, whereas SLU models utilize global features. The authors propose masking certain utterances to break local dependencies and explore the development of a learnable mask generator.

**Strengths:**

1. The proposed method outperforms existing baselines significantly in terms of computational efficiency and memory usage, which are based on disentangled encoders.
2. The evaluation includes both black-box adversaries, who only have access to the masked signal, and white-box adversaries, who also have access to the encoder.
3. The paper is well-organized, making it an easy read.

**Weaknesses:**

1. My primary concern is the paper's limitation to passive adversaries. An active adversary might develop models specifically to reconstruct the raw signal from the masked signal. Such a threat is not investigated in this paper.
2. While the disruption of local dependencies does not affect the performance of user intent classification, this approach may not apply to all scenarios. Certain tasks, such as detecting key phrases or specific commands such as “set my clock at 10 AM” rely on local utterances.

**Questions:**

1. Could the authors elaborate on the potential ASR accuracy when facing an active adversary that trains a model to reconstruct raw signals from the masked signals?
2. Is it possible to visualize some masks generated by the trained mask generator to verify that they effectively disrupt local utterances, aligning with the main motivation of the study?

---

> ### Author Rebuttal · Authors · 2024-08-07
>
> Thanks for comments and questions. Here are the answers to them:
>
> > *W1. Concern about limitation to passive adversaries.*
>
> We clarify it and add evaluation of defending against reconstruction adversaries in **Q1**.
>
> > *W2.  This approach may not apply to all scenarios*
>
> We admit that *detecting key phrases or specific commands* is not the focus of this work.
> In most of the experiments, we treat these as sensitive words to preserve.
>
> However, we want to emphasize that long-term dependent intent classification remains a major objective of SLU understanding literature [4] and has a wide range of applications.
> For example, it can be used for command recognition, sensor activation monitoring, and more.
> These applications are particularly relevant for resource-constrained devices.
>
> We have added this clarification into the updated manuscript.
>
> > *Q1. Could the authors elaborate on the potential ASR accuracy when facing an active adversary that trains a model to reconstruct raw signals from the masked signals?*
>
> Certainly, we conducted further experiments and offer additional clarification to better elaborate on the system's performance when facing active adversaries. We also have Figure 1 in the uploaded pdf to better illustrate the different attack scenarios.
>
> First of all, we want to clarify a possible misunderstanding: the white-box adversary, `Whisper (White-box)`, not only has access to the encoder but also collects enormous data from malicious users to learn how to predict the lost manuscript.
> Thus, it might be considered a form of active predictive attack.
> The raw description can be found in lines 251-253 of the manuscript:
>
> ```
> Whisper (White-box) then utilizes this collected data from malicious users to adapt the pre-trained Whisper.medium.en model to the specific masking pattern.
> ```
>
> Besides, we have implemented two more active reconstruction adversaries and demonstrated our efficiency in defending against them.
> - `U-Net` is a traditional inpainting model based on convolutional U-Net structure, commonly used in literature to reconstruct missing audio signals [1,2].
> We utilize the SLURP training set and their masked counterparts to train the inpainting model from scratch to reconstruct the missing audio.
> - `CQT-Diff` is a neural diffusion model with an invertible Constant-Q Transform (CQT) to leverage pitch-equivariant symmetries [3], allowing it to effectively reconstruct audio without retraining.
>
> The reconstructed audio is sent to Whisper for automatic recognition. The visualizations of reconstructed waveforms have also been included in the global PDF for your reference.
>
> The updated evaluation results under attacks are summarized in the table below.
> |                   | **PlainText** | **Azure** | **Whisper** | **U-Net** | **CQTdiff** | **Whisper (white box)** |
> |:-----------------:|:-------------:|:---------:|:-----------:|:---------:|:-----------:|:-----------------------:|
> | **WER-SLU (%)**   |      14.7     |    81.6   |     78.6    |    82.5   |     74.3    |           67.3          |
> | **WER-ASR (%)**   |      12.3     |    71.6   |     68.1    |    71.4   |     65.9    |           64.4          |
>
> *Table: Potential attack Word Error Rate (WER) of different attack scenarios. PlainText means AllOffloaded without any privacy protection.*
>
> As shown in the table, `U-Net` can rarely reconstruct the masked audio.
> Even worse, it introduces incorrect noisy signals, degrading the attack success rate.
> `CQT-Diff` inpainting can fill the missing waveforms but cannot successfully reconstruct the content because it is designed to reconstruct background music, such as piano concertos.
> SLU audio, which includes human intent and conversation, is difficult to reconstruct.
>
> `Whisper (white-box)` is fine-tuned to predict the missing transcript directly and shows the best attack performance among all attack methods.
> However, with our system enabled, it still results in over a 60\% word error rate, which is 50\% higher than without our preservation.
> Thus, our system can successfully protect the private content in command audio under a wide range of attacks, including passive and active attacks.
>
> > *Q2. Is it possible to visualize some masks generated by the trained mask generator to verify that they effectively disrupt local utterances, aligning with the main motivation of the study?*
>
> Sure! We have visualized some masks generated by the trained mask generator.
> The figure has been added to the uploaded PDF in the global rebutta.
>
> It can be seen that the mask generator can dispatch suitable mask granularity to proper speech granularity to some extent.
> With more semantics utterance around, the mask becomes more meticulous, with the slices being distributed accordingly.
>
> References are listed in the global rebuttal.

---

> > ### Comment · Reviewer_YHpd · 2024-08-12
> >
> > Thank you for the reply and for adding the new experiments. They address some of my concerns and I have increased my score to 5.

---

> > > ### Author Response · Authors · 2024-08-12
> > >
> > > Dear reviewer,
> > >
> > > We sincerely appreciate your thorough review of our rebuttal and the increased score. We kindly request further guidance on any remaining concerns you may have.
> > >
> > > We are more than willing to provide additional clarification to enhance your understanding and satisfaction with our work.
> > >
> > > Thank you once again for your recognition and assistance.
> > >
> > > Best regards,
> > >
> > > Authors

---

> ### Author Response · Authors · 2024-08-12
>
> Dear Reviewer,
>
> Thank you again for reviewing our manuscript. We have tried our best to address your concerns and questions (see our rebuttal in the top-level comment and above), and revised our paper by following suggestions from all reviewers.
>
> Additionally, we have included more references to underscore the importance of our focus on long-dependent intent classification in SLU literature [1-9].
>
> Please kindly let us know if you have any follow-up questions or areas needing further clarification. Your insights are valuable to us, and we stand ready to provide any additional information that could be helpful.
>
> Best regards,
>
> Authors
>
> ```
> [1]. Rongxiang Wang and Felix Xiaozhu Lin. Turbocharge Deep Speech Understanding on the Edge. to appear at Proc. ACM Int. Conf. Mobile Computing and Networking (MobiCom), 2024.
> [2]. Soham Deshmukh, Benjamin Elizalde, Rita Singh, and Huaming Wang. Pengi: An audio language model for audio tasks. Advances in Neural Information Processing Systems (NeurIPS), 2023.
> [3]. Jixuan Wang, Martin Radfar, Kai Wei and Clement  Chung. End-to-end spoken language understanding using joint CTC loss and self-supervised, pretrained acoustic encoder. IEEE International Conference on Acoustics, Speech and Signal Processing (ICASSP), 2023.
> [4]. Bhuvan Agrawal, Markus Muller, Samridhi Choudhary, Martin Radfar, Athanasios Mouchtaris, Ross McGowan, Nathan Susanj, and Siegfried Kunzmann. Tie your embeddings down: Cross-modal latent spaces for end-to-end spoken language understanding. IEEE International Conference on Acoustics, Speech and Signal Processing (ICASSP), 2022.
> [5]. Libo Qin, Tianbao Xie, Wanxiang Che and Ting Liu. Proceedings of the Thirtieth International Joint Conference on Artificial Intelligence (IJCAI), 2021.
> [6]. Emanuele Bastianelli, Andrea Vanzo, Pawel Swietojanski, and Verena Rieser. SLURP: A Spoken Language Understanding Resource Package. In Proceedings of the 2020 Conference on Empirical Methods in Natural Language Processing (EMNLP), 2020.
> [7]. Shinji Watanabe, Takaaki Hori, Suyoun Kim, John R. Hershey, and Tomoki Hayashi. Hybrid ctc/attention architecture for end-to-end speech recognition. IEEE Journal of Selected Topics in Signal Processing, 11(8):1240–1253, 2017.
> [8]. Jan K Chorowski, Dzmitry Bahdanau, Dmitriy Serdyuk, Kyunghyun Cho, and Yoshua Bengio. Attention-based models for speech recognition. In C. Cortes, N. Lawrence, D. Lee, M. Sugiyama, and R. Garnett, editors, Advances in Neural Information Processing Systems, volume 28. Curran Associates, Inc., 2015.
> [9]. Renato De Mori. Spoken language understanding: A survey. In 2007 IEEE Workshop on Automatic Speech Recognition & Understanding (ASRU), pages 365–376. IEEE, 2007.
> ```

---

### Official Review · Reviewer_ubkn · 2024-07-08

**Soundness:** 3
**Presentation:** 2
**Contribution:** 3
**Rating:** 5
**Confidence:** 2

**Summary:**

The paper presents SILENCE, a method designed to address privacy concerns in cloud-based speech services by selectively obscuring short-term dependencies in speech signals. This technique preserves speech understanding functionality while protecting sensitive information. Implemented on the STM32H7 microcontroller, SILENCE significantly enhances speed and memory efficiency compared to existing solutions, effectively protecting privacy against various attack scenarios. Key contributions of the paper include an innovative encoder design based on asymmetric dependencies, the integration of automated masking configuration, and the demonstration of SILENCE's practical feasibility on low-resource devices.

**Strengths:**

- Innovative use of differential mask generators for privacy-preserving speech understanding on constrained devices.
- Thorough experimental validation demonstrating significant performance and efficiency gains.
- Clear explanations of the key concepts and methodology, though some technical details could be more detailed.

**Weaknesses:**

- While the paper evaluates the system under black-box and white-box attacks, it does not thoroughly address the robustness of the masking approach against adaptive adversaries who might use advanced techniques to reverse-engineer the mask patterns.

- Masking short-term dependent frames to protect privacy could lead to losing crucial phonetic information necessary for accurate SLU. There is a lack of detailed analysis on how the granularity of the mask affects both privacy and SLU performance at different levels of speech granularity.

- The method relies heavily on the assumption that SLU tasks are primarily long-term dependent while ASR tasks are short-term dependent. This binary distinction might not hold for all SLU tasks, especially those involving contextually rich and intricate utterances.

- The experiments are primarily conducted on a single dataset. Evaluating the method on a broader range of datasets would better demonstrate its generalizability and robustness.

**Questions:**

- How does SILENCE perform against adaptive adversaries using advanced reverse-engineering techniques?
- How does mask granularity affect both privacy protection and SLU performance at various levels of speech granularity?
- How does SILENCE handle SLU tasks that require short-term dependencies?
- How does SILENCE perform on additional datasets to ensure its generalizability and robustness across different speech scenarios?

**Limitations:**

The authors adequately addressed the limitations and potential negative societal impact of their work.

---

> ### Author Rebuttal · Authors · 2024-08-07
>
> Thank you for your insightful comments and questions. Here are the answers to the questions:
>
> > *Q1. How does SILENCE perform against adaptive adversaries using advanced reverse-engineering techniques?*
>
> Our system can still preserve content privacy under advanced reconstruction attacks, achieving over 64\% recognized word error rates.
>
> We have implemented two more active reconstruction adversaries and demonstrated our efficiency in defending against them.
> - `U-Net` is a traditional inpainting model based on convolutional U-Net structure, commonly used in literature to reconstruct missing audio signals [1,2].
> We utilize the SLURP training set and their masked counterparts to train the inpainting model from scratch to reconstruct the missing audio.
> - `CQT-Diff` is a neural diffusion model with an invertible Constant-Q Transform (CQT) to leverage pitch-equivariant symmetries [3], allowing it to effectively reconstruct audio without retraining.
>
> The reconstructed audio is sent to Whisper for automatic recognition. The visualizations of reconstructed waveforms have also been included in the global PDF for your reference.
>
> The updated evaluation results under attacks are summarized in the table below.
> |                   | **PlainText** | **Azure** | **Whisper** | **U-Net** | **CQTdiff** | **Whisper (white box)** |
> |:-----------------:|:-------------:|:---------:|:-----------:|:---------:|:-----------:|:-----------------------:|
> | **WER-SLU (%)**   |      14.7     |    81.6   |     78.6    |    82.5   |     74.3    |           67.3          |
> | **WER-ASR (%)**   |      12.3     |    71.6   |     68.1    |    71.4   |     65.9    |           64.4          |
>
> *Table: Potential attack Word Error Rate (WER) of different attack scenarios. PlainText means AllOffloaded without any privacy protection.*
>
> As shown in the table, `U-Net` can rarely reconstruct the masked audio.
> Even worse, it introduces incorrect noisy signals, degrading the attack success rate.
> `CQT-Diff` inpainting can fill the missing waveforms but cannot successfully reconstruct the content because it is designed to reconstruct background music, such as piano concertos.
> SLU audio, which includes human intent and conversation, is difficult to reconstruct.
>
> > *Q2. How does mask granularity affect both privacy protection and SLU performance at various levels of speech granularity?*
>
> We included two more fine-grained speech understanding tasks: action and the combined intent (scenario_action) recognition.
> For your reference, there are 18 different scenarios and 46 defined actions, resulting in 828 possible combinations for intend.
>
> |                   | AllOffloaded | VAE  | PPSLU | OnDevice | Ours |
> |:------------:|:------------:|:----:|:-----:|:--------:|:----:|
> | ACC-Scenario (\%) | 88.2         | 72.8 | 73.9  | 88.2     | 80.2 |
> | ACC-Action (%)    | 77.1         | /    | /     | 77.1     | 76.4 |
> | ACC-Intent (%)    | 83.3         | /    | /     | 83.3     | 76.8 |
> | WER-SLU (%)       | 14.7         | /    | /     | 100      | 68.6 |
> | WER-ASR (\%)      | 12.3         | 69.3 | 75.3  | 100      | 68.1 |
>
> *Table: Comparison between Privacy-preservation and SLU performance at different speech granularities. ‘/’ means not supported. Local leaks no words as nothing is uploaded.*
>
> It can be seen that our method can recognize speech intent at different granularities.
> For example, we can correctly recognize 76.8\% of the combined intent.
>
> In comparison, disentanglement-based methods need to re-entangle representations for different semantic granularities.
> Thus, the classifier used for scenario classification cannot be applied to other intents, and these methods are not designed to preserve the sensitive information within command audios.
> This emphasises a significant advantage of our approach, as it does not require retraining the model for different intent granularities.
>
> > *Q3. How does SILENCE handle SLU tasks that require short-term dependencies?*
>
> We admit that our work primarily focuses on long-term dependent SLU classification tasks.
> In most of our experiments, we treat entities requiring short-term dependencies as sensitive words to preserve.
>
> However, we want to emphasize that long-term dependent intent recognition is a major objective of SLU understanding literature [4] and has a wide range of applications.
> For example, it can be used for command recognition, sensor activation monitoring, and more.
> These applications are particularly relevant for resource-constrained devices.
>
> > *Q4. How does SILENCE perform on additional datasets to ensure its generalizability and robustness across different speech scenarios?*
>
> We conducted further experiments on the Fluent Speech Commands (FSC) dataset [5], another widely used dataset for spoken language understanding research.
> The FSC dataset includes 97 speakers and 30,043 relevant utterances.
> We split the data, using 20\% for testing and the remaining 80\% for training.
> The results are shown below.
>
> |               | AllOffloaded | VAE  | PPSLU | Local | Random | SILENCE |
> |:-------------:|:------------:|:----:|:-----:|:-----:|:------:|:-------:|
> | ACC-SLU (\%)  | 99.7         | 98.3 | 99.2  | 99.7  | 86.4   | 99.1    |
> | WER\-ASR (\%) | 1.2          | 65.5 | 78.5  | 100   | 76.6   | 81.4    |
>
> *Table: Evaluation of privacy preservation and SLU performance on FSC dataset.*
>
> Our system demonstrates accurate intent understanding (more than 99\%, similar to all the baselines) and effectively defends against sensitive word recognition attacks (achieving more than 80\% WER, outperforming all disentanglement-based protections).
> Additionally, our method significantly outperforms existing baselines in terms of computational efficiency and memory usage, allowing for a wider range of deployment scenarios.
>
> References are listed in the global rebuttal.

---

> ### Author Response · Authors · 2024-08-12
>
> Dear Reviewer,
>
> Thank you again for reviewing our manuscript. We have tried our best to address your concerns and questions (see our rebuttal in the top-level comment and above), and revised our paper by following suggestions from all reviewers.
>
> Additionally, we have included more references to underscore the importance of our focus on long-dependent intent classification in SLU literature [1-9].
>
> Please kindly let us know if you have any follow-up questions or areas needing further clarification. Your insights are valuable to us, and we stand ready to provide any additional information that could be helpful.
>
> Best regards,
>
> Authors
>
> ```
> [1]. Rongxiang Wang and Felix Xiaozhu Lin. Turbocharge Deep Speech Understanding on the Edge. to appear at Proc. ACM Int. Conf. Mobile Computing and Networking (MobiCom), 2024.
> [2]. Soham Deshmukh, Benjamin Elizalde, Rita Singh, and Huaming Wang. Pengi: An audio language model for audio tasks. Advances in Neural Information Processing Systems (NeurIPS), 2023.
> [3]. Jixuan Wang, Martin Radfar, Kai Wei and Clement  Chung. End-to-end spoken language understanding using joint CTC loss and self-supervised, pretrained acoustic encoder. IEEE International Conference on Acoustics, Speech and Signal Processing (ICASSP), 2023.
> [4]. Bhuvan Agrawal, Markus Muller, Samridhi Choudhary, Martin Radfar, Athanasios Mouchtaris, Ross McGowan, Nathan Susanj, and Siegfried Kunzmann. Tie your embeddings down: Cross-modal latent spaces for end-to-end spoken language understanding. IEEE International Conference on Acoustics, Speech and Signal Processing (ICASSP), 2022.
> [5]. Libo Qin, Tianbao Xie, Wanxiang Che and Ting Liu. Proceedings of the Thirtieth International Joint Conference on Artificial Intelligence (IJCAI), 2021.
> [6]. Emanuele Bastianelli, Andrea Vanzo, Pawel Swietojanski, and Verena Rieser. SLURP: A Spoken Language Understanding Resource Package. In Proceedings of the 2020 Conference on Empirical Methods in Natural Language Processing (EMNLP), 2020.
> [7]. Shinji Watanabe, Takaaki Hori, Suyoun Kim, John R. Hershey, and Tomoki Hayashi. Hybrid ctc/attention architecture for end-to-end speech recognition. IEEE Journal of Selected Topics in Signal Processing, 11(8):1240–1253, 2017.
> [8]. Jan K Chorowski, Dzmitry Bahdanau, Dmitriy Serdyuk, Kyunghyun Cho, and Yoshua Bengio. Attention-based models for speech recognition. In C. Cortes, N. Lawrence, D. Lee, M. Sugiyama, and R. Garnett, editors, Advances in Neural Information Processing Systems, volume 28. Curran Associates, Inc., 2015.
> [9]. Renato De Mori. Spoken language understanding: A survey. In 2007 IEEE Workshop on Automatic Speech Recognition & Understanding (ASRU), pages 365–376. IEEE, 2007.
> ```

---

### Official Review · Reviewer_BMwq · 2024-07-10

**Soundness:** 2
**Presentation:** 3
**Contribution:** 4
**Rating:** 6
**Confidence:** 4

**Summary:**

This paper presents a lightweight speech intent understanding paradigm for wimpy devices, with heavy concern on the privacy. It targets the recently-popular disentanglement-based approaches for speech processing. It reached a decent balance between efficiency and privacy preservation, and make the SLU system work on wimpy devices.

**Strengths:**

1. The paper addresses real-time problem with its own novelty, which is quite rare for recent papers who mostly prefer large-scale solution with massive and massive data. The novelty here is thus quite high.
2. The paper has a lot of measurements and concerns about hardware-wise perspective, which echoes (1).

**Weaknesses:**

Since this is more like a engineer-oriented work, the reviewer does not have strong problem or weakness from technical point of view.

But the reviewer does have a big problem on the architecture of SILENCE. Seems like from Figure 5, it still relies on cloud technology to maintain part of the framework. However, the wimpy devices sometimes (if not mostly) will meet offline conditions. In such case, the paper does not have an intention of making the system work.

**Questions:**

1. Compared to conventional SLU systems, do you think end-to-end systems preserve the nature of enhanced privacy? Do you think your algorithm will work for both the end-to-end system and the conventional modularized system?
2. The reviewer has problem with the definition of wimpy devices, since raspberry PI shall not be considered as "wimpy". Chips such as ARM cortex V7 has much lower memory and a lot of start-up are still using it for their offline solutions. The reviewer thinks "resource-constrained conditions" might be a better term. Of if there is official or your clear definition of "wimpy" devices with detailed stats, the reviewer is happy to keep head down and learn.
3. The reviewer are interested into the proposed framework with conventional methods.

**Limitations:**

The reviewer does not see any limitation or ethical concern about the paper.

---

> ### Author Rebuttal · Authors · 2024-08-07
>
> Thank you for your insightful comments, and we greatly appreciate your high praise regarding the novelty of our work.
> We hope that our rebuttal response will further enhance the soundness of our research.
> Below are the answers to your concern and questions:
>
> > *W1. The paper does not have an intention of making the system work under offline conditions.*
>
> We completely agree that offline conditions occur periodically for resource-constrained devices.
> However, we want to emphasize that our system can be easily integrated into an orchestration of small on-device SLU models and robust cloud models.
> This orchestration has been officially adopted by many off-the-shelf products, such as Apple Intelligence in iOS 18.
>
> Our system remains indispensable in such circumstances because small on-device SLU models may not generate satisfactory intent understanding due to their restricted model size.
> Even when on-device SLU models produce correct intent understanding, they cannot always operate due to limited device energy.
> As a result, online procedures are still the main components of current SLU solutions.
> The on-device functionality can be used as an alternative in offline conditions.
> With our system, the cloud-based SLU component is both privacy-preserving and efficient.
>
> > *Q1.1. Compared to conventional SLU systems, do you think end-to-end systems preserve the nature of enhanced privacy?*
>
> We think the end-to-end system preserves enhanced privacy during both training and inference.
> During training, the end-to-end system does not require raw transcripts to improve SLU performance, as illustrated in Figure 5 of the paper.
> During inference, skipping the generation of intermediate transcripts can also enhance privacy and directly improve the final SLU performance.
>
> > *Q1.2. Do you think your algorithm will work for both the end-to-end system and the conventional modularized system?*
>
> Yes, our algorithm can correctly detect SLU intent in both the end-to-end system and the conventional modularized system.
> The conventional modularized system can recognize intent with nearly 90\% accuracy because it uses extra intermediate correct transcripts to enhance the first ASR module.
> Added implementation and experiments results are detailed in the **Q5**.
>
> > *Q2. The reviewer has problem with the definition of wimpy devices, since raspberry PI shall not be considered as "wimpy". Chips such as ARM cortex V7 has much lower memory and a lot of start-up are still using it for their offline solutions. The reviewer thinks "resource-constrained conditions" might be a better term. Of if there is official or your clear definition of "wimpy" devices with detailed stats, the reviewer is happy to keep head down and learn.*
>
> We agree that the term "resource-constrained conditions" is more explicit to a wider range of readers.
> Our initial use of "wimpy" was intended to include the following types of devices:
> - Devices that do not have enough power to run the model, such as the STM32H7 microcontroller.
> - Devices that have enough power to run the model but can save energy by offloading computations to the cloud, e.g., Raspberry Pi.  Reasons to include those devices have been discussed in **W1**.
>
> Thank you again for helping us select a clearer term.
>
> > *Q3. The reviewer are interested into the proposed framework with conventional methods.*
>
> We applied our algorithm to conventional modularized SLU models.
> The experimental results demonstrate that when both the ASR and NLU modules are fine-tuned as required, the conventional modularized SLU model can recognize intent correctly when fed with masked audio.
> The detailed results are summarized in the table below:
>
> |                   | Plaintext |   VAE   |  PPSLU  | NLU only (Ours) | Decoupled SLU (Ours) | E2E SLU (Ours) |
> |:-----------------:|:---------:|:-------:|:-------:|:------------------:|:-----------------------:|:-----------------:|
> | **SLU-ACC (%)**       |   87.2    |  72.5   |  74.5   |        12.6        |          89.1           |        81.1       |
>
> *Table: System performance on conventional modularized SLU. Plaintext equals to AllOffloaded or OnDevice.*
>
> Here, the ASR model of the modularized SLU is Whisper.medium.en, and the NLU model is with two LSTM layers.
> For the column labeled "NLU only," the ASR module uses pre-trained weights downloaded directly from Hugging Face without any fine-tuning.
> During training, the real transcript is used as input, and audio intent is used as a label to reduce the loss.
> During inference, the masked audio is processed through the pre-trained ASR and fine-tuned NLU model sequentially to obtain the final intents.
> However, it cannot achieve accurate intent understanding because the intermediate transcripts are corrupted.
>
> For the column labeled "Decoupled SLU," the ASR module is also fine-tuned to better match the masked audio and real transcript as the first step.
> As a result, the final intent recognition is hugely improved because it obtains real transcripts from users and preserves the intend information beforehand.

---

> ### Author Response · Authors · 2024-08-12
>
> Dear Reviewer,
>
> Thank you again for reviewing our manuscript. We have tried our best to address your concerns and questions (see our rebuttal in the top-level comment and above), and revised our paper by following suggestions from all reviewers.
>
> Additionally, we have included more references to underscore the importance of our focus on long-dependent intent classification in SLU literature [1-9].
>
> Please kindly let us know if you have any follow-up questions or areas needing further clarification. Your insights are valuable to us, and we stand ready to provide any additional information that could be helpful.
>
> Best regards,
>
> Authors
>
> ```
> [1]. Rongxiang Wang and Felix Xiaozhu Lin. Turbocharge Deep Speech Understanding on the Edge. to appear at Proc. ACM Int. Conf. Mobile Computing and Networking (MobiCom), 2024.
> [2]. Soham Deshmukh, Benjamin Elizalde, Rita Singh, and Huaming Wang. Pengi: An audio language model for audio tasks. Advances in Neural Information Processing Systems (NeurIPS), 2023.
> [3]. Jixuan Wang, Martin Radfar, Kai Wei and Clement  Chung. End-to-end spoken language understanding using joint CTC loss and self-supervised, pretrained acoustic encoder. IEEE International Conference on Acoustics, Speech and Signal Processing (ICASSP), 2023.
> [4]. Bhuvan Agrawal, Markus Muller, Samridhi Choudhary, Martin Radfar, Athanasios Mouchtaris, Ross McGowan, Nathan Susanj, and Siegfried Kunzmann. Tie your embeddings down: Cross-modal latent spaces for end-to-end spoken language understanding. IEEE International Conference on Acoustics, Speech and Signal Processing (ICASSP), 2022.
> [5]. Libo Qin, Tianbao Xie, Wanxiang Che and Ting Liu. Proceedings of the Thirtieth International Joint Conference on Artificial Intelligence (IJCAI), 2021.
> [6]. Emanuele Bastianelli, Andrea Vanzo, Pawel Swietojanski, and Verena Rieser. SLURP: A Spoken Language Understanding Resource Package. In Proceedings of the 2020 Conference on Empirical Methods in Natural Language Processing (EMNLP), 2020.
> [7]. Shinji Watanabe, Takaaki Hori, Suyoun Kim, John R. Hershey, and Tomoki Hayashi. Hybrid ctc/attention architecture for end-to-end speech recognition. IEEE Journal of Selected Topics in Signal Processing, 11(8):1240–1253, 2017.
> [8]. Jan K Chorowski, Dzmitry Bahdanau, Dmitriy Serdyuk, Kyunghyun Cho, and Yoshua Bengio. Attention-based models for speech recognition. In C. Cortes, N. Lawrence, D. Lee, M. Sugiyama, and R. Garnett, editors, Advances in Neural Information Processing Systems, volume 28. Curran Associates, Inc., 2015.
> [9]. Renato De Mori. Spoken language understanding: A survey. In 2007 IEEE Workshop on Automatic Speech Recognition & Understanding (ASRU), pages 365–376. IEEE, 2007.
> ```

---

> ### Author Response · Authors · 2024-08-12
> **Correction to Reference in Response to Q1.2**
>
> Dear Reviewer,
>
> We apologize for the typo in our previous communication regarding the reference for “Added implementation and experiments results of conventional modularized system.” The details are provided in **Q3** instead of **Q5**.
>
> Best regards,
>
> Authors

---

### Official Review · Reviewer_rbWb · 2024-07-13

**Soundness:** 3
**Presentation:** 3
**Contribution:** 3
**Rating:** 5
**Confidence:** 2

**Summary:**

This paper proposed a private speech processing system that selectively obscures short-term details to reduce privacy leakage in cloud-based Spoken Language Understanding (SLU). A differential mask generator is learned to automatically mask out portions of audio signals along with online cloud inference with the generated masks. Empirical results show that the proposed system offers comparable speech understanding performance and privacy protection capacity with high memory efficiency.

**Strengths:**

\+ Tackled a practical and crucial topic of cloud-based SLU privacy protection.

\+ detailed system illustration and rationale interpretation.

\+ Comprehensive experiment details.

**Weaknesses:**

Could provide more background info for attack scenarios.

**Questions:**

What types of devices can be defined as 'wimpy' devices?

---

> ### Author Rebuttal · Authors · 2024-08-07
>
> Thank you for your comments and questions. Here are the answers to the points:
>
> > *W1. Could provide more background info for attack scenarios.*
>
> Attacks may involve passive and active adversaries attempting to automatically recognize private entities in uploaded command audio files.
> Passive adversaries use off-the-shelf automatic speech recognition (ASR) models to conduct recognition on the masked audio.
> Active adversaries may collect data from malicious users to build reverse or predictive models that can either reconstruct the raw audio waveform or directly reconstruct the correct final transcript.
>
> **We have provided an illustration of several attack scenarios in Figure 1 of the global rebuttal PDF.
> Please refer to it for a more detailed description.**
>
> > *Q1. What types of devices can be defined as 'wimpy' devices?*
>
> Wimpy devices refer to devices that either lack the power to run robust transformer-based spoken language understanding (SLU) models, such as the STM32H7 microcontroller, or those IoT devices that have the capability to run these models but have limited power, such as the Raspberry Pi 4.
> Offloading computation to the cloud can help these devices save energy.
> As suggested by Reviewer BMwq, we will change the term to "resource-constrained conditions" to improve clarity.

---

> ### Author Response · Authors · 2024-08-12
>
> Dear Reviewer,
>
> Thank you again for reviewing our manuscript. We have tried our best to address your concerns and questions (see our rebuttal in the top-level comment and above), and revised our paper by following suggestions from all reviewers.
>
> Additionally, we have included more references to underscore the importance of our focus on long-dependent intent classification in SLU literature [1-9].
>
> Please kindly let us know if you have any follow-up questions or areas needing further clarification. Your insights are valuable to us, and we stand ready to provide any additional information that could be helpful.
>
> Best regards,
>
> Authors
>
> ```
> [1]. Rongxiang Wang and Felix Xiaozhu Lin. Turbocharge Deep Speech Understanding on the Edge. to appear at Proc. ACM Int. Conf. Mobile Computing and Networking (MobiCom), 2024.
> [2]. Soham Deshmukh, Benjamin Elizalde, Rita Singh, and Huaming Wang. Pengi: An audio language model for audio tasks. Advances in Neural Information Processing Systems (NeurIPS), 2023.
> [3]. Jixuan Wang, Martin Radfar, Kai Wei and Clement  Chung. End-to-end spoken language understanding using joint CTC loss and self-supervised, pretrained acoustic encoder. IEEE International Conference on Acoustics, Speech and Signal Processing (ICASSP), 2023.
> [4]. Bhuvan Agrawal, Markus Muller, Samridhi Choudhary, Martin Radfar, Athanasios Mouchtaris, Ross McGowan, Nathan Susanj, and Siegfried Kunzmann. Tie your embeddings down: Cross-modal latent spaces for end-to-end spoken language understanding. IEEE International Conference on Acoustics, Speech and Signal Processing (ICASSP), 2022.
> [5]. Libo Qin, Tianbao Xie, Wanxiang Che and Ting Liu. Proceedings of the Thirtieth International Joint Conference on Artificial Intelligence (IJCAI), 2021.
> [6]. Emanuele Bastianelli, Andrea Vanzo, Pawel Swietojanski, and Verena Rieser. SLURP: A Spoken Language Understanding Resource Package. In Proceedings of the 2020 Conference on Empirical Methods in Natural Language Processing (EMNLP), 2020.
> [7]. Shinji Watanabe, Takaaki Hori, Suyoun Kim, John R. Hershey, and Tomoki Hayashi. Hybrid ctc/attention architecture for end-to-end speech recognition. IEEE Journal of Selected Topics in Signal Processing, 11(8):1240–1253, 2017.
> [8]. Jan K Chorowski, Dzmitry Bahdanau, Dmitriy Serdyuk, Kyunghyun Cho, and Yoshua Bengio. Attention-based models for speech recognition. In C. Cortes, N. Lawrence, D. Lee, M. Sugiyama, and R. Garnett, editors, Advances in Neural Information Processing Systems, volume 28. Curran Associates, Inc., 2015.
> [9]. Renato De Mori. Spoken language understanding: A survey. In 2007 IEEE Workshop on Automatic Speech Recognition & Understanding (ASRU), pages 365–376. IEEE, 2007.
> ```

---

### Author Rebuttal · Authors · 2024-08-07

Dear Reviewers,

**Please see the attached PDF for a one-page summary with an illustrative description of different attack scenarios, visualizations of generated masks and reconstructed waveforms, and additional experiment results against advanced active reconstruction attacks.**

We would like to thank all reviewers for providing constructive feedback that helped us improve the paper. We are encouraged that reviews think our paper:

- “addresses real-time problem with its own novelty, which is quite rare for recent papers who mostly prefer large-scale solution with massive and massive data. The novelty here is thus quite high.” (Reviewer BMwq)
- provides “comprehensive experiment details” (Reviewer rbWb) with “significant performance and efficiency gains” (Reviewer ubkn) and “outperforms existing baselines significantly” (Reviewer YHpd)
- provides “detailed system illustration and rationale interpretation” (Reviewer rbWb), “making it an easy read.” (Reviewer YHpd)

We have been working diligently on improving the paper in several aspects, addressing your concerns and problems. Below, we summarise the changes that we have made in an updated draft.

**1. Experiment results against advanced reconstruction attacks**
- We first illustrated passive and active adversaries in the uploaded pdf, including the advanced reconstruction attacks.
- We implemented two advanced inpainting techniques to reconstruct the missing waveforms. (1) U-Net: a traditional inpainting method based on convolutional U-Net [1, 2]. (2) CQT-Diff: a neural diffusion model preconditioned with an invertible Constant-Q Transform (CQT) [3].
- The reconstructed waveforms are visualised in the attached PDF to see their effects.
- We conduct experiments to show that our system can still preserve content privacy under those advanced reconstruction attacks, with over 64% recognised word error rates. Detailed results are summarised in the attached PDF.

**2. Evaluation of the additional dataset**
- We have conducted further experiments on Fluent Speech Commands (FSC) dataset [5], another widely used dataset for spoken language understanding research.
- Our system can still give accurate intent understanding (more than 99%, similar level with all the baselines) and defend against the sensitive word recognition attack (more than 80% WER, better than all the disentanglement-based protections).

|               | AllOffloaded | VAE  | PPSLU | Local | Random | SILENCE |
|:-------------:|:------------:|:----:|:-----:|:-----:|:------:|:-------:|
| **ACC-SLU (\%)** | 99.7         | 98.3 | 99.2  | 99.7  | 86.4   | 99.1    |
| **WER-ASR (\%)** | 1.2          | 65.5 | 78.5  | 100   | 76.6   | 81.4    |

*Table: Evaluation of privacy preservation and SLU performance on FSC dataset.*

**3. Evaluation of the conventional modularised SLU**
- We implemented a conventional modularised SLU system with Whisper.medium.en as the ASR modular and two LSTM layers as the NLU modular.
- We conducted experiments on both `NLU only` and `Decouple SLU` settings to demonstrate that our system can still generate accurate intent understanding if we fine-tune the ASR modular.

|                   | Plaintext |   VAE   |  PPSLU  | NLU only (Ours) | Decoupled SLU (Ours) | E2E SLU (Ours) |
|:-----------------:|:---------:|:-------:|:-------:|:------------------:|:-----------------------:|:-----------------:|
| **SLU-ACC(%)**       |   87.2    |  72.5   |  74.5   |        12.6        |          89.1           |        81.1       |

*Table: System performance on conventional modularized SLU. Plaintext equals to AllOffloaded.*

**4. Evaluation on different speech granularity**
- We select two more speech granularities including Action (46 classes) and Intent (828 classes) classification.
- We conducted experiments to show that our system can understand the speech intent on different speech granularities.

|                   | AllOffloaded | VAE  | PPSLU | OnDevice | Ours |
|:------------:|:------------:|:----:|:-----:|:--------:|:----:|
| **ACC-Scenario (\%)** | 88.2         | 72.8 | 73.9  | 88.2     | 80.2 |
| **ACC-Action (%)**    | 77.1         | /    | /     | 77.1     | 76.4 |
| **ACC-Intent (%)**    | 83.3         | /    | /     | 83.3     | 76.8 |
| **WER-SLU (%)**       | 14.7         | /    | /     | 100      | 68.6 |
| **WER-ASR (\%)**      | 12.3         | 69.3 | 75.3  | 100      | 68.1 |

*Table: Comparison between privacy-preservation capacity and speech understanding performance at different speech granularities. ‘/’ means not supported. OnDevice leaks no words as nothing is uploaded.*

**5. More helpful visualisation and analysis**

We added visualisation of generated masks in the uploaded pdf to verify that they effectively disrupt certain local utterances.

**6. More clear statements and insightful discussion**
- We added a clear definition of the wimpy devices.
- We added a discussion about the integration with small on-device SLU models to address the occasional offline conditions.
- We clarified that detecting short-dependent key phrases or specific commands is not the focus of this work. But we also emphasised the importance of long-dependent intend classification. It is currently the main objective of SLU understanding literature [4] and has a wide range of application scenarios.

Please see our reviewer-specific feedback for more information.

---

[1]. Masking and inpainting: A two-stage speech enhancement approach for low SNR and non-stationary noise, ICASSP’23

[2]. Deep speech inpainting of time-frequency masks, interspeech’19

[3]. CQTDiff: Solving audio inverse problems with a diffusion model, ICASSP’23

[4]. A Survey on Spoken Language Understanding: Recent Advances and New Frontiers, IJCAI’21.

[5]. Fluent Speech Commands: A dataset for spoken language understanding research, Fluent.ai.

---

### Decision · Program_Chairs · 2024-09-25

**Decision:**

Accept (poster)

**Comment:**

The paper presents a  speech processing system that selectively obscures short-term details to reduce privacy leakage.  The proposed solution, a differential mask, is is based on the observation that speech understanding hinges on long-term dependency knowledge of the entire utterance, in contrast to privacy-sensitive elements that are short-term dependent. The authors propose a differential mask generator that automatically masks out portions of audio signals during online cloud inference. Empirical results show that the proposed system offers comparable speech understanding performance and privacy protection capacity with high memory efficiency.

The extensive additional experiments that that the authors have conducted during the rebuttal phase should be included in the paper. These include the illustrative description of different attack scenarios, visualizations of generated masks and reconstructed waveforms, and additional experiment results against advanced active reconstruction attacks, as well as the evaluation on the Fluent Speech Commands (FSC) dataset, the evaluation of the conventional modularised SLU and the different speech granularity.

The authors are also encouraged to replace "wimpy" with "resource-constrained", as it would be easier for the community to infer what they mean. Regardless, they should explain precisely and clearly what resources are deemed constrained, to what extent (i.e., in a quantifiable fashion), and why devices such as, e.g., Rasberry PIs pass/fail the definition test. Finally, active adversaries should also be explicitly addressed, both in the paper's presentation but also through the inclusion of the conducted additional experiment results against advanced active reconstruction attacks.